# Human–Chelonian Bond in Italy: An Exploratory Study of Pet Turtle and Tortoise Ownership

**DOI:** 10.3390/vetsci12080773

**Published:** 2025-08-18

**Authors:** Carmen Borrelli, Alexandre Azevedo, Manuel Magalhães-Sant’Ana, Chiara Mariti

**Affiliations:** 1Department of Veterinary Sciences, University of Pisa, 5614 Pisa, Italy; carmen.borrelli@phd.unipi.it; 2Vasco da Gama Research Center (CIVG), Vasco da Gama University School (EUVG), Avenida José R. Sousa Fernandes 197, Lordemão, 3020-210 Coimbra, Portugal; ax.c.azevedo@gmail.com; 3Centro de Investigação Interdisciplinar em Sanidade Animal (CIISA), Faculdade de Medicina Veterinária, Universidade de Lisboa, 1300-477 Lisbon, Portugal; mdsantana@fmv.ulisboa.pt; 4Laboratório Associado para Ciência Animal e Veterinária (AL4AnimalS), 1300-477 Lisbon, Portugal

**Keywords:** chelonians, human–animal bond, reptile ownership, pets, pet guardian, tortoises, *Testudo hermanni*

## Abstract

Reptiles such as turtles and tortoises are becoming increasingly popular as pets in Europe, including Italy. However, research regarding the relationship between guardians and their reptiles is scarce. This study explored why Italian guardians choose to keep chelonians (the scientific group that includes turtles or tortoises) as pets, with a particular focus on their emotional bond. Ninety-one questionnaires were collected from across the country, with most participants keeping Hermann’s tortoises outdoors. Nearly half of the respondents described their bond with their chelonians using emotional terms such as love, fascination, or affection. People who kept their turtles indoors were more likely to consider them family members than those who kept them outside. Compared to dogs and cats, chelonians were generally perceived as less emotionally attached to their guardians. Still, many owners valued their reptiles for their uniqueness, low maintenance, and fascination. These findings suggest that even non-conventional pets like reptiles can hold an important emotional role in people’s lives, though the nature of this bond may differ from that with traditional pets. Understanding these relationships can help improve both animal welfare and the well-being of their guardians.

## 1. Introduction

Across Europe, the ownership of reptiles as pets has been on the rise, reflecting a growing fascination with exotic animals [1]. A report by the Federation of European Pet Food Industry estimated that as of 2022, there were approximately 12 million terraria species living in European households [2]. Countries such as Germany, France, Italy, Russia, and the United Kingdom are among the leading markets for exotic pets, including reptiles [2]. This trend is partly driven by the appeal of reptiles’ unique characteristics, their perceived lower maintenance costs compared to conventional pets, and the expanding availability of reptile care products and information [3].

In Italy, the growing interest in reptiles is reflected by the approximately 1.4 million reptiles kept as pets, representing a significant share of the country’s 65 million total pets [4]. The Assalco-Zoomark [4] report highlights this surge, noting that while reptiles are still fewer in number compared to dogs and cats, they represent a key part of Italy’s pet landscape. The growing popularity of reptiles in Italy might be driven by the curiosity about exotic species and the desire for pets perceived as easier to care for in an urban environment. This trend also raises important considerations for pet care and welfare, as reptiles have specific needs in terms of habitat, nutrition, and health, which differ greatly from those of more common pets [5,6,7].

The human–animal bond has been extensively studied, particularly in relation to companion animals such as dogs and cats, with researchers investigating the physical, psychological, and social benefits associated with pet ownership [8,9]. Reptiles, in contrast, are generally perceived as less emotionally engaging pets, largely due to their non-mammalian biology and distinct behavioral patterns [10,11]. However, in recent years, some studies began to question this perception, suggesting that reptiles’ guardians can also develop strong emotional connections with their pets [12,13]. Studies on the human–animal bond often rely on standardized and validated questionnaires as the Lexington Attachment to Pet Scale (LAPS) [14] and the Cat/Dog Owner Relationship Scale (C/DORS) [15,16,17]. However, little is known about how these frameworks apply to non-conventional pets such as reptiles, which present distinct characteristics in terms of interaction and care. Another theoretical framework often used to study the human–animal bond is the attachment theory by John Bowlby [18], which explains how two individuals (usually a child and his caregiver) can be emotionally tied in space and time. In recent decades, many studies have shown that this kind of bond is even possible with pets, especially dogs [19,20,21,22] and possibly cats, although results for the latter are often inconsistent [23,24,25]. Regarding reptiles, however, the application of the attachment theory is challenging. On the one hand, reptiles do not usually exhibit the typical behavior indicative of attachment towards their guardians (like seeking proximity or protest during involuntary separation). On the other hand, the nature of interaction between guardians and reptiles differs significantly from that with mammals.

Despite the widespread use of reptiles as pets in many Western countries, the scientific literature on the human–reptile relationship is notoriously scarce (especially regarding the Italian and Southern European context), and usually focused on characteristics of pet reptile guardians, husbandry, and possible risks of zoonosis (e.g., Salmonella) [26,27,28].

Azevedo and colleagues [12] conducted one of the few studies focused on the human–reptile bond, examining the motivations behind reptile ownership and the emotional factors that influence these relationships in the Portuguese population. The findings suggested that reptile guardians perceived their pets as sentient beings and family members, forming strong bonds with them. Additionally, they found that the motivations for acquiring a reptile involve similar feelings to those previously reported for mammalian pets. Building on the work of Azevedo et al. [12], this study aims to explore the human–reptile bond, specifically the human–chelonian bond, within the Italian population. The focus is on understanding the husbandry and care conditions of chelonians, motivations behind their acquisition, the emotional dynamics involved in this relationship, and identifying the demographic factors that may influence the strength and nature of this bond. The findings from this research are expected to enhance our understanding of reptile–human interactions and support the development of guidelines for improving the welfare of both chelonian pets and their guardians.

## 2. Materials and Methods

### 2.1. Ethics Statement

This study received the approval of the Bioethics Committee of the University of Pisa (approval n. 5/2022), being a part of a wider research project investigating the human–pet relationship using questionnaires. Participation was voluntary, and the questionnaire was anonymous. Participants had to be at least 18 years old and own a chelonid. Respondents were required to provide informed consent before proceeding with the online questionnaire. They were required to read an informative sheet where they were briefed on the objectives of the research, as well as the procedures for data collection, management, and dissemination.

For the sake of clarity, the authors would like to emphasize that, in accordance with ethical considerations supported by international organizations (for example, ASPCA [29]), the terms “guardian” has been used instead of “owner” throughout the whole text. Nevertheless, since the term “guardianship” is still not widely adopted, we chose to use “ownership” when necessary.

### 2.2. Survey Design

The questionnaire from Azevedo et al. [12] was translated into Italian and, with the help of a veterinarian working predominantly with exotic animals not involved in the study, adapted to ensure it was suitable for the Italian context.

The questionnaire was divided into five sections (the full survey is provided in Appendix A). The first section focused on the animal’s demographics (scientific name, age, sex, when and where the reptile was acquired, and if the participants had additional pets). The second section regarded information about husbandry and care. The third section gathered information about the human–animal relationship, particularly about social interaction and type of bond (family member, friend, pet, burden, or other). In the fourth section, participants were asked to interpret the causes behind ten previously validated reptile behaviors (for detailed information cf. Azevedo et al. [12]). For the first four sections, the questions were mandatory, and an “Other/Don’t Know” option was available. Section five, gathering participants’ demographic information (age, gender, housing, region, education, marital status, and occupation), was optional; two open-ended questions to explore the motivation for acquiring (mandatory) and keeping (optional) a pet chelonian were additionally asked. Compared to the original survey by Azevedo et al. [12], a question was added regarding the perceived attachment score of the chelonian to their guardians from 1 (very little) to 5 (very much); for those participants who also owned a dog or cat, the same question was asked referring to the latter.

The questionnaire was disseminated online through the Google Forms platform. The link to the survey was spread among students and personnel of the Department of Veterinary Sciences, University of Pisa (Pisa, Italy), through the official websites and social pages of the university (Instagram and Facebook). In addition, clients of a veterinary clinic in Pisa (Centro Veterinario Exotic, Pisa, Italy) were voluntarily recruited through a QR code displayed in the clinic’s waiting room, which provided direct access to the survey. The form was further disseminated on social media pages and/or groups on Facebook dedicated to reptiles and additionally using snowball sampling. The data were collected between 29 April and 9 October 2023.

### 2.3. Statistical Analysis

Descriptive statistics were run with the software IBM SPSS statistics ^®^ (version 25.0; IBM Corp., Armonk, NY, USA) and included frequencies and barplots of the responses. Non-parametric (Wilcoxon signed rank test) analyses were run to compare the perceived attachment score of reptiles compared to traditional companion animals (dogs and cats). Inferential analyses were performed with the statistical software R (version 4.4.2) [30] with a significance level of α = 0.05. A logistic regression model was used to investigate whether the odds of reporting pet chelonians as a family member (vs. non-family member) was influenced by their species, the way they were acquired, whether they had access to indoors or lived exclusively outdoors, and respondent characteristics (gender, education, profession, civil state, type of house, and level of urbanization). For the purpose of logistic regression (but not descriptive analyses), missing data were generated by multivariate imputation by chained equations using the package “mice” [31], and multicollinearity issues were tested for using the variance inflation factor (VIF) with a cut-off value of 2, using the function car::vif [32]. Variable levels with zero or very few observations were dealt with by grouping them with other levels based on biological meaning. For example, in the case of bond, the levels “animal” and “friend” were grouped to obtain a variable with only two levels: “family-member” and “non-family-member”. Similarly, in the case of species, observations were grouped into three levels: “*Testudo hermanni*” (*n* = 64), “other tortoise species” (*n* = 6), and “aquatic or semi-aquatic turtle” (*n* = 4). The variable *urbanization* was also recoded by combining the categories “isolated” and “village” into a single “rural” category, while “town” and “city” were retained as separate levels. The *acquisition* variable was recoded into three categories, “purchase,” “gift,” and “other,” with the latter combining infrequent responses such as “adopted” and “found” (*n* = 5). Model selection based on the Akaike Information Criteria was used to determine which of the candidate predictors improved model parsimony, using the step AIC function of the MASS package [33]. The results are presented as odds ratios, 95% confidence intervals, and *p*-values (OR, 95% CI, *p*-value). However, *p*-values are included solely for completeness and transparency and are not used as the primary basis for interpreting the results. The interpretation focuses on effect size, directionality, and the plausibility of associations in the context of an exploratory study.

### 2.4. Qualitative Analysis

Written responses explaining why participants chose to acquire and keep a reptile as a pet were extracted and imported into NVIVO software, Release 1.7.2 (QSR International, February 2024). Deductive coding was applied based on the categories identified by Azevedo et al. [12], namely affective states, reasons for acquiring, motivations for keeping reptiles, and negative aspects of reptile ownership. The initial coding round was conducted by the third author (M.M.-S.), a male veterinarian specialized in animal welfare and experienced in thematic analysis, following the same analytical approach used by Azevedo et al. [12]. As a constructivist reliability measure, a second round of coding was collaboratively performed by C.B., A.A., and M.M.-S. with the aim of maximizing coverage and ensuring consensus. The Italian language was translated into English by the authors, who included one native Italian speaker (C.B.) and one native English speaker (A.A.), with careful attention to preserving the original meaning. All data were anonymized prior to analysis to ensure confidentiality.

## 3. Results

### 3.1. Demographics

In total, 91 chelonid guardians replied to the questionnaire, 67 women (73.6%) and 22 men (24.2%) (two preferred not to say), and the mean age was 44 years old (min 18, max 70; SD: 11.5). Geographically, most came from the center (42.9%) and northwest (30.8%) parts of Italy. Participants predominantly lived in cities (41.8%) or towns (38.5%), and 87.9% of respondents had a house with an outdoor space (private garden). Regarding education, 31.9% of participants completed secondary education, 37.4% had a university degree, and 23.1% had post-degree education (e.g., PhD., specialization). Professionally, 37.4% were employees, 19.8% were self-employed, and the remaining 9.9% did not have an occupation (either student, retired, or unemployed). One in five participants (19.8%) worked with animals. Most of the participants (88% *n* = 76) accessed the survey through social media and/or groups on Facebook dedicated to reptiles. Appendix A provides the full demographic information of participants.

Concerning the species of pet chelonians, 85.7% of participants owned Hermann’s tortoises (*Testudo hermanni*), 4.4% owned aquatic turtles, and 9.9% owned other species of tortoises (all species are described in Table 1). Most of these animals (86.8%) lived outdoors, while the remaining 13.2% lived indoors or in mixed environments. The majority of participants received their chelonian as a gift (50.5%) or purchased them (31.9%).

Among the species reported, *Testudo hermanni* is the only chelonian native to Italy. The remaining species—including *Testudo graeca*, *T. marginata*, and *T. horsfieldii*—are non-native Mediterranean or Central Asian species that are regularly kept under CITES regulation. Some aquatic species, such as *Trachemys scripta*, are classified as invasive and are subject to trade restrictions under European regulation. Nevertheless, some individuals may still be kept legally by owners who acquired them before the restrictions came into force.

### 3.2. Assessment of Factors Influencing the Bond Type

Before modeling, we examined the unadjusted distribution of bond types across demographic and environmental factors. These descriptive frequencies, including percentages and 95% confidence intervals, are presented in Table 2.

Based on the descriptive findings in Table 2, we explored whether the selected demographic and contextual variables could explain the variation in bond type using logistic regression. Given the limited sample size and the exploratory nature of the study, we adopted a cautious model-selection strategy, using stepwise AIC-based reduction and minimizing overfitting by limiting model complexity. Table 3 summarizes the candidate predictors considered for model inclusion.

Among the candidate variables for model construction (Table 3), only environment was retained in the final model following AIC-based stepwise selection. Other variables were considered but did not improve model fit (in blue in Figure 1; for model selection results see Appendix A).

Pet chelonians living outdoors had higher odds of being reported as non-family members compared to those with indoor access (OR = 4.90, 95% CI = 1.34–23.41, *p* = 0.02, Figure 1 and Figure 2, Table 4). Although this estimate suggests a possible association, the wide confidence interval reflects substantial uncertainty, and this result should be interpreted cautiously in the context of an exploratory model. The results were confirmed using the non-parametric chi-square [34] and Fisher’s exact [35] tests to assess the relation between categorical predictors and bond type and Spearman’s correlation [36] to assess the relation between numerical predictors and bond type. The results indicated a significant relationship (χ^2^(1, N = 91) = 4.42, *p* = 0.036) between environment and bond type, and no relation was observed with any of the other predictors.

Of all of the guardians (*n* = 39/91), 43% considered their chelonian a family member, while 26.4% saw it as an animal and 11.0% as a friend; the remaining 19.8% did not fit into these categories (“none of the above mentioned”). Looking at species distribution in *Testudo hermanni*, 42.3% were considered family members, 12.8% a friend, and 25.6% an animal. Regarding other species, aquatic turtles were most frequently considered family members (75.0%), while just 25.0% of guardians considered them “an animal” and none “a friend”. Other terrestrial tortoises were equally considered a family member, an animal, or did not fit in any of the categories (33.3% each). Regarding the environment in which each species lives, the majority of *Testudo hermanni* were kept outdoors (91.1%), with only 5.1% living in mixed environments and 3.8% living exclusively indoors. Other terrestrial tortoises followed a similar trend, with 88.9% kept outdoors and only 11.1% in mixed environments. Aquatic turtles instead were more likely to be housed indoors (75%), with 25% living in a mixed environment.

### 3.3. Perceived Attachment to Guardians

Seventy participants replied to the question regarding perceived attachment to guardians for both chelonian and traditional (dogs and cats) pets. The results highlighted that dogs/cats were perceived as more attached to their guardians (*p* < 0.001; median chelonian: 2.00; median pet: 5.00; Figure 3).

The same 70 guardians also replied to the question of whether they consider their chelonian or cat/dog a family member. Of these, 44.3% (*n* = 31/70) of chelonian guardians viewed their animal as part of the family, compared to 55.7% (*n* = 39/70) who did not. In contrast, 92.9% (*n* = 65/70) of these respondents considered their cat/dog pet to be a family member.

### 3.4. Qualitative Data

Ninety-five percent of participants (*n* = 86/91) explained why they decided to acquire and keep a turtle as a pet. One response was excluded from the analysis because its context was unclear. Among these, 38 participants (45%) relied on terms denoting an emotional connection with their animals, including fondness, love, fascination, and passion (Table 5). One 20-year-old female urban student owning Hermann tortoises (*Testudo hermanni*) expressed how “*they are fascinating animals that, like any pet, need care and attention. I like to observe them and learn from them*”.

About half of the respondents to the open-ended questions (49.4%; *n* = 42) elaborated on the reasons for acquiring chelonians. Most animals were received as gifts (40.5%; *n* = 17), followed by adoption or rescue (23.8%; *n* = 10), inheritance (14.3%; *n* = 6), and other reasons (16.7%; *n* = 7), including by chance (*n* = 3). An urban 53-year-old female employee shared the experience of finding a male Greek tortoise (*Testudo graeca*) on the street (“*He was injured and I treated him and then he stayed with me*”). A 27-year-old sub-urban female adopted a Hermann’s tortoise “*because the former guardians were moving to a house without a garden*”. Past experiences with chelonians, especially during childhood, emerged as a notable factor, accounting for eight cases (9.4%; *n* = 8). One 37-year-old woman living in an isolated house mentioned how “*I gave one [Testudo hermanni] to my mother who had one when she was little and I fell in love with it too*”.

The identified motivations for keeping chelonians (after acquisition) include entertainment/leisure (12.9%; *n* = 11), convenience (11.8%; *n* = 10), duty of care (8.2%; *n* = 7), and companionship (3.5%; *n* = 3). In terms of entertainment, tortoises and turtles were described as interesting, fascinating, relaxing, and good to observe and learn from. Examples include a rural female, aged 52, who justified keeping Hermann’s tortoises “*because they are endangered animals and represent a window into prehistory*”. Convenience includes the perception that chelonians are “*independent and that do not need interaction with humans to live well*”, as noted by a young sub-urban male student. Convenience is also framed as an alternative to cats and dogs (2.4%; *n* = 2), exemplified by a middle-aged female urban employee who explained: “*My daughter wanted a pet but my husband is allergic to dogs. We decided on [Hermann’s] tortoises*”. The concept of duty of care denotes those guardians who describe the keeping of their chelonians as a responsibility or an obligation that was often imposed on them, namely those animals acquired by adoption, rescue, or inheritance. In contrast, three guardians described the ownership of chelonians in terms of companionship and sociability.

Some negative aspects of chelonian keeping have also been identified (4.7%; *n* = 4). These include the lack of control over the ownership of the animal (“*My husband decided. I didn’t want to. Then the tortoise became mine*”). Another aspect that was mentioned regards the challenges of providing adequate standards of care, as illustrated by the following quote from a guardian (female, urban, 27) who received a Hermann’s tortoise as a gift:

“*Because I thought he needed little space, in reality knowing his behavior better I realized that he would need much more space than I can offer him, in fact I intend to bring him to my parents’ garden*.”

## 4. Discussion

This study investigated the human–chelonian relationship within a convenience sample of the Italian population. It focused on understanding the husbandry and care, type of bond, and motivations for acquiring chelonians as pets. The sample had a high prevalence of Hermann’s tortoises (*Testudo hermanni*, 85.7%) compared to other terrestrial or aquatic chelonian species. This represents a higher proportion of native species when compared with the Portuguese sample, which contained only 3 (out of 220) animals that could be considered autochthonous [12]. Additionally, 86.8% of the chelonians in the sample were kept outdoors. Importantly, *Testudo hermanni* is widely distributed throughout most of Italy [37] and can therefore be considered as adapted to the climatic conditions experienced in the outdoor environment, where most of the animals are kept. Together with the increased complexity of outdoor environments compared to terraria, this suggests that the *Testudo hermanni* specimens in this sample may experience better welfare compared to the non-native species in the Portuguese study. On the other hand, it may raise concerns regarding the origin of these animals and the implications for the species’ conservation. The collection of specimens for pet trade, as well as genetic pollution and diseases from released pet tortoises, are listed among the major threats to the species [38].

In our sample, nearly half of the guardians considered their chelonian a family member (43%). The only factor associated with bond type in our exploratory model was the housing environment. Respondents that housed their chelonians in outdoor environments were almost five times more likely to describe them as non-family members (OR = 4.90, 95% CI = 1.34–23.41). While the effect estimate was supported by a relatively wide confidence interval, this association was also visible at the descriptive level (see Table 4). Species that were mainly kept indoors tended to be considered family members more often (e.g., aquatic turtles) compared with those living outdoors (e.g., terrestrial ones). Chelonians kept indoors may interact more frequently with their guardians (through observation, handling, and routine care), thus fostering a stronger sense of companionship. In contrast, outdoor housing may limit both the frequency and intensity of human–animal interactions, thereby reducing the likelihood of these animals being perceived as family members [39]. In addition, for some species kept outdoors, environmental constraints such as the need for large enclosures, exposure to natural elements, and seasonal behaviors like hibernation may further restrict opportunities for contact and emotional bonding. Cultural beliefs may also help explain this pattern. As noted by Janovcová et al. [10], human attitudes toward reptiles are often shaped by fear and disgust as well as by aesthetic preferences. Although chelonians are usually perceived more favorably than other reptiles such as snakes, they are still placed culturally outside the category of traditional companion animals such as dogs and cats. In our sample, where the majority of chelonians were housed outdoors with limited opportunities for daily contact, these cultural perceptions may further reinforce their role as animals to be observed rather than as true family members.

Other demographic or environmental factors such as education level, urbanization, or gender showed no significant relationships with the bond type. While inconclusive, the lack of significant results could suggest that the bond with reptiles is less influenced by socio-demographic characteristics compared to that with dogs and cats, where such variables have shown strong effects [16]. However, given the limited sample size, these results should be interpreted with caution, as a more representative sample might reveal some of these more subtle relationships.

In addition, chelonians were less frequently considered family members than dogs and cats among people owning both. These findings align with previous studies using the Cat/Dog Owner Relationship Scale (C/DORS) to evaluate the pet–owner relationship, which consistently demonstrated high attachment levels among dog and cat guardians [16,17]. C/DORS measures attachment through scales focused on interaction, emotional closeness, and perceived companionship, dimensions where mammals, particularly dogs and cats, traditionally score highly. Reptiles, and particularly chelonians, may not have the same characteristics. The human–pet bond that exists between dogs/cats and their guardians is frequently explained by theories like the attachment bond [18,40] and social support [41]. Both rely on humans physically or socially interacting with pets. For reptile guardians, as shown in this study and supported by Azevedo et al. [12], the emotional connection appears more nuanced. The relationship with chelonians, though meaningful, lacks the same interactive qualities, likely influencing the lower scores in perceived attachment scores. Unlike dogs and cats, chelonians often tolerate less tactile interaction and are less expressive, factors that may shape how they are perceived as family members. In addition, most chelonians in this study were kept outdoors, and this might have influenced their perceived bond. Currently, most dogs and cats are kept indoors or in a mixed environment, with more opportunities to directly interact with their guardian and form a stronger bond [42]. Furthermore, it is important to acknowledge that attachment theory has been developed primarily in the context of mammals and may not be directly applicable to reptiles. In species like dogs, the attachment bond is facilitated by the presence of social behaviors, parental dependency in the first period of life, and mutual bonding cues [19,20,21,22]. In contrast, chelonians have different biological characteristics, with hatchlings being fully independent from birth and receiving no parental care. In this sense, from an evolutionary and ecological perspective, they might not be predisposed to form attachment bonds in the same way mammals are. Consequently, this may be also an explanation of why guardians perceived their chelonians to be less attached than their dog or cat, as found in this study.

Another factor that could influence the bond is neoteny. In mammals, juvenile traits such as large eyes, round faces, and small noses trigger parental care instincts [43]. While neoteny is well-documented in more conventional pets like dogs and cats [44], it is unclear whether it plays a role in reptile ownership and the reasons for adopting. Haddon [11] suggests that some reptiles, such as lizards, may be perceived as “cute” due to their rounded features, whereas snakes and turtles generally lack these neotenic characteristics. This could partly explain why, in our study, chelonians are less frequently perceived as family members compared to dogs and cats.

Importantly, the perception of a weaker bond does not imply the absence of emotional connection. As suggested by Macauley and Chur-Hansen [39], relationships with non-conventional animals may involve alternative forms of connection that do not depend on interaction but more on fascination, caregiving, and a sense of purpose and routine. For instance, the passive observation of animals kept in their enclosures, which should recreate their natural environments, might facilitate a sense of connectedness with nature (possibly acting as Animal-Nature-Based Solutions [45]) for the guardians and improve their psychological health.

Our qualitative findings regarding the emotional connection between guardians and their chelonians challenge the traditional view of reptiles as less emotionally engaging pets [11] and support more recent research suggesting that reptile guardians can develop meaningful bonds with their pets [12,14]. Like the findings of the Portuguese study [12], a significant number of Italian respondents (45%) resorted to at least one affective state to explain why they decided to acquire and keep a pet chelonian. These affective states—namely, love, fascination, and passion—convey a strong emotional connection with these animals. Although the proportion of respondents making use of these terms is slightly lower in the present study than in the Portuguese one (45% against 54%), those denoting fascination by the turtles remained virtually unchanged (10.5% against 10.6%).

The qualitative findings also reinforce the connection between childhood experiences and the ownership of reptiles, an association that has been previously described in the literature [46]. This suggests potential long-term educational implications. Early exposure to these animals might influence future pet choices and contribute to broader conservation awareness, given that several species of chelonians are endangered in their natural habitats.

Most chelonians in our sample were kept for entertainment or convenience. In contrast, the most cited reasons for acquiring a pet dog or cat are companionship and the desire to increase physical activity [47,48]. Additionally, many guardians report choosing a pet for the social support they provide [47]. The prevalence of unplanned acquisitions in our study—through gifts, inheritance, or rescue—raises important challenges for animal welfare and public education for responsible pet keeping. Contrary to popular belief, chelonians are not low-maintenance pets and require complex and rich environments with adequate diets, temperature, and lighting [6]. While the emotional connection between guardians and their chelonians is evident, the unplanned nature of many acquisitions may lead to inadequate preparation for the complex requirements these animals need. This finding aligns with previous research suggesting that impulse purchases and the gifting of exotic pets often result in suboptimal care conditions [3].

Several limitations of the study should be considered. The convenience nature of our sample may have introduced a degree of self-selection bias, potentially resulting in the over-representation of more engaged and passionate chelonian guardians. Additionally, the sample size was relatively small and geographically concentrated in central and northern Italy, thus limiting the generalizability of our findings. Future research would benefit from a larger, more geographically diverse sample and could include longitudinal studies to examine how the human–chelonian bond develops over time.

## 5. Conclusions

Amidst an increasing trend of non-conventional pet ownership, characterizing human–animal relations with these pets is required to inform policy and awareness efforts. This study confirms the existence of a bond between guardians and chelonians, though with somewhat distinct characteristics compared to conventional pets (dogs and cats). Our findings suggest that the bond among chelonian guardians is based on a relationship less oriented toward interaction and emotional connection while still holding personal significance. Specifically, chelonian guardians appear to base their connection more on values such as entertainment and convenience rather than companionship and mutual interaction. This may prompt a re-evaluation of how the human–animal bond is defined in the context of non-conventional pets. Among the examined variables, living outdoors was the only significant predictor of a non-family-member-type bond. The lack of other statistically significant predictors suggests that demographic or situational factors may have less influence on the human–chelonian bond than they do for conventional pets, but further research with more robust sampling is necessary to confirm this.

## Figures and Tables

**Figure 1 vetsci-12-00773-f001:**
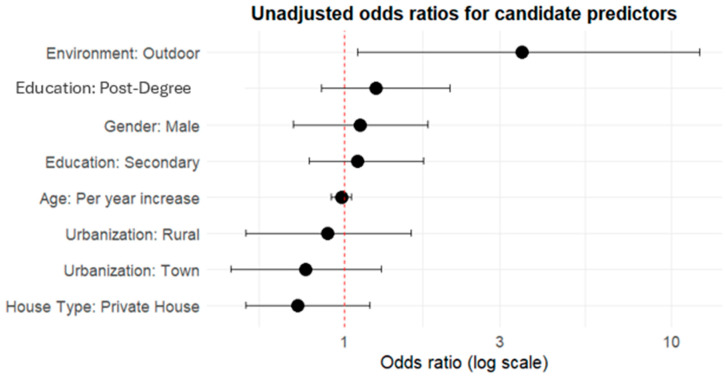
Unadjusted odds ratios for candidate predictors of non-family-member bond formation. This forest plot illustrates unadjusted odds ratios and 95% confidence intervals from logistic regression models, each including only one predictor variable at a time.

**Figure 2 vetsci-12-00773-f002:**
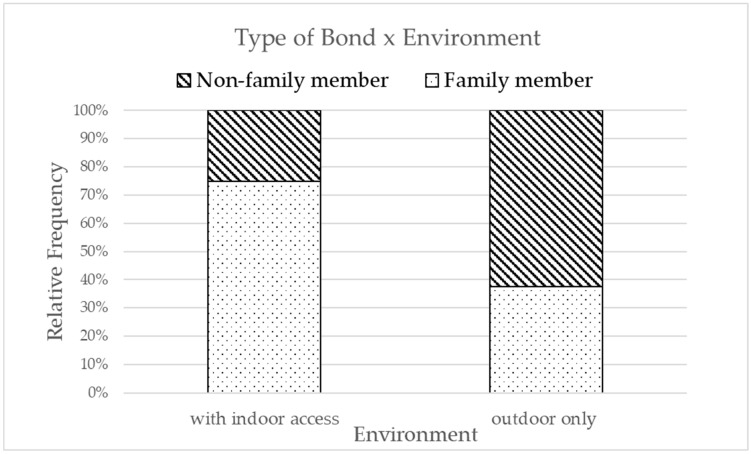
Distribution of the type of bond (family member vs. non-family member) based on the environment.

**Figure 3 vetsci-12-00773-f003:**
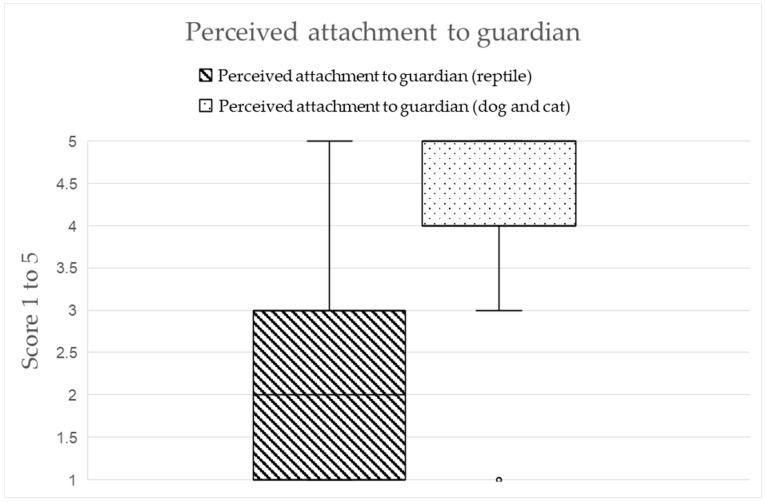
Comparison between perceived attachment of reptiles vs. conventional pets to their guardian. Score goes from 1 = very little to 5 = very much.

**Table 1 vetsci-12-00773-t001:** List of chelonid species.

Genus	Frequencies	Percentage (%)
*Testudo hermanni*	78	85.7
*Testudo graeca*	3	3.3
*Testudo marginata*	3	3.3
*Testudo horsfieldii*	2	2.2
*Trachemys* spp.	2	2.2
*Graptemys* spp.	1	1.1
*Sternotherus* spp.	1	1.1
*Testudo elegans*	1	1.1

**Table 2 vetsci-12-00773-t002:** Frequency of reported bond type (“family-member” vs. “non-family-member”) by environment, species group, and gender of guardian. Percentages and 95% confidence intervals (CIs) reflect the proportion of each bond type within levels of each variable.

Variable	Level	Family Member (%, 95% CI)	Non-Family Member (%, 95% CI)
Environment	Outdoor	30 (38.5%) [27.7–49.3]	48 (61.5%) [50.7–72.3]
	Indoor/Mixed	9 (81.8%) [59.0–104.6]	2 (18.2%) [−4.6–41.0]
Species or group	*T. hermanni*	33 (42.9%) [31.8–53.9]	44 (57.1%) [46.1–68.2]
	Aquatic turtle	3 (75.0%) [32.6–117.4]	1 (25.0%) [−17.4–67.4]
	Other tortoise	3 (37.5%) [4.0–71.0]	5 (62.5%) [29.0–96.0]
Guardian’s gender	Female	36 (49.3%) [37.8–60.8]	37 (50.7%) [39.2–62.2]
	Male	6 (30.0%) [7.1–52.9]	14 (70.0%) [47.1–92.9]

Note: This table refers to 89 out of 91 participants; 2 respondents were excluded due to missing or incomplete information for one or more of the variables shown (bond type, environment, species, or gender).

**Table 3 vetsci-12-00773-t003:** Candidate variables used in the logistic regression model assessing the predictors of bond type. Variables retained in the final model after stepwise model selection are highlighted with *.

Variable	Levels
Bond type (dependent variable)	Family member or non-family member
Species	Testudo hermanni; other tortoise species; or aquatic or semi-aquatic turtle
Acquisition	Purchase; gift; or other
Environment *	Outdoor only or indoor or mixed
Gender	Female or male
Region	Northeast; northwest; center or south and islands
Urbanization	City; town; or rural
Education	Secondary; degree; or post-degree
Profession	Animal-related; employee; self-employed; or student, pensioner, or unemployed
Civil state	Married; cohabiting; or single
House	House with private garden; or apartment or house with shared garden

**Table 4 vetsci-12-00773-t004:** Unadjusted and adjusted odds ratios. This table presents the unadjusted and adjusted odds ratios, confidence intervals, and *p*-values for logistic regression models examining the relationship between candidate variables and the likelihood of forming a non-family-member bond with a reptile pet. The adjusted odds ratio for “environment” corresponds to the final model, which retained only this variable after AIC-based model selection. Other variables were not included in the adjusted model due to insufficient contribution to model parsimony. Note: *p*-values are reported for completeness but are not used to determine variable importance.

Variable	Categories	Unadjusted Odds Ratio	95% CI (Lower)	95% CI (Upper)	Adjusted Odds Ratio	95% CI (Lower)	95% CI (Upper)	*p*-Value
Environment	Indoor/Mixed (Ref)	—	—	—	—	—	—	—
	Outdoor	**3.50**	**1.10**	**12.20**	**4.90**	**1.34**	**23.41**	**0.024**
Age	Per year Increase	0.98	0.91	1.05	—	—	—	—
Gender	Female (Ref)	—	—	—	—	—	—	—
	Male	1.12	0.70	1.80	—	—	—	—
Urbanization	City (Ref)	—	—	—	—	—	—	—
	Rural	0.89	0.50	1.60	—	—	—	—
	Town	0.76	0.45	1.30	—	—	—	—
Education	Degree (Ref)	—	—	—	—	—	—	—
	Post-Degree	1.25	0.85	2.10	—	—	—	—
	Secondary	1.10	0.78	1.75	—	—	—	—
House Type	Apartment/Shared (Ref)	—	—	—	—	—	—	—
	Private House	0.72	0.50	1.20	—	—	—	—
Intercept	—	—	—	—	−1.10	0.07	1.12	0.099

**Table 5 vetsci-12-00773-t005:** Terms used by reptile guardians in Italy to describe affective states towards their animals. The “*n*” denotes the frequency each term was used by the 38 respondents. Four respondents used two terms.

Affective State (in English)	Original Italian Term	*n*	Percentage (Out of 85 Responses)
Fondness (to like)	Piacere, mi piacciono	11	12.9
Love	Amo, adoro	13	15.3
Fascination	Affascinanti	9	10.6
Passion	Passione, appassionano	4	4.7
Other: attractive, inspiring, irresistible	Altro: attratta, mi ispirava, irresistibile	5	5.9

## Data Availability

Full database of the study can be found at https://osf.io/x32ap/ (accessed on 23 June 2025).

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
