# Peer review of "Human–Chelonian Bond in Italy: An Exploratory Study of Pet Turtle and Tortoise Ownership"

_vetsci, 2025, doi:10.3390/vetsci12080773_

Round 1
Reviewer 1 Report
Comments and Suggestions for Authors
This manuscript addresses an increasingly relevant topic in the field of human–animal interactions, particularly regarding the emotional and practical dimensions of exotic pet ownership. The focus on chelonian guardianship within the Italian context provides a novel contribution to the existing literature. The use of a mixed-methods approach is a notable strength, enabling the authors to triangulate quantitative data with meaningful qualitative insights. The inclusion of an emotional language analysis also adds originality and depth to the study.
To further enhance the clarity, methodological transparency and impact of the paper, I respectfully suggest the following suggestions:
- contextualisation of literature: expanding the background section could better situate the work within the broader literature on human–reptile relationships, highlighting existing gaps in the Italian or Southern European context;
- theoretical framing: a brief mention of relevant theoretical frameworks, such as attachment theory or concepts from multispecies ethnography, may help to ground the interpretations of emotional bonding;
- sample limitations: although the sample size provides useful insights, the recruitment via online survey may have introduced a degree of self-selection bias. It would be valuable to acknowledge this more clearly in the limitations section;
- demographic variables: if data are available, it could be useful providing more detailed demographic information (e.g., gender, region, rural vs. urban context). This may help readers better understand potential influences on perceptions and bonding;
- interpretation of some findings: the data showing that outdoor-housed chelonians are less likely to be considered family members is compelling and it could be beneficial to deeper discuss possible underlying factors, such as differences in daily interaction, cultural beliefs or environmental constraints;
- clarification of statistical analysis: it is not well clear if the logistic regression model was adjusted for additional covariates or if the reported odds ratio is unadjusted;
- qualitative data presentation: the qualitative responses are rich and important, anyway including anonymised illustrative quotes in the main text or as supplementary material would support the thematic interpretations and improve reader engagement;
- methodological transparency: a clearer explanation of the qualitative analysis process, including coding strategy, whether inductive or deductive and whether inter-coder reliability was assessed, would enhance the study’s rigour;
- terminology consistency: the decision to refer to pet keepers as “guardians,” which is a thoughtful and ethically loaded choice, may be improved to briefly explain this terminology and ensure consistent use throughout the manuscript.
Overall, this is a well-designed and insightful study that brings attention to an under-researched aspect of human–animal relationships. I believe the manuscript has the potential to make a valuable contribution to the field once the above points are addressed.

Author Response
This manuscript addresses an increasingly relevant topic in the field of human–animal interactions, particularly regarding the emotional and practical dimensions of exotic pet ownership. The focus on chelonian guardianship within the Italian context provides a novel contribution to the existing literature. The use of a mixed-methods approach is a notable strength, enabling the authors to triangulate quantitative data with meaningful qualitative insights. The inclusion of an emotional language analysis also adds originality and depth to the study.
Answer to general comment: We wish to thank the reviewer for these very positive comments and the following suggestions.
To further enhance the clarity, methodological transparency and impact of the paper, I respectfully suggest the following suggestions:
Comment R1.1: contextualisation of literature: expanding the background section could better situate the work within the broader literature on human–reptile relationships, highlighting existing gaps in the Italian or Southern European context;
Answer R1.1: Thank you for the suggestion; we have now added some literature, highlighting the gaps in the Italian and Southern European context at lines 79-82
Comment R1.2: Theoretical framing: a brief mention of relevant theoretical frameworks, such as attachment theory or concepts from multispecies ethnography, may help to ground the interpretations of emotional bonding.
Answer R1.2:
Thank you for this suggestion. We have mentioned these in the discussion at lines 396-431, in order to explain the differences between the bond and interactions people can establish with traditional pets compared to reptiles. Nevertheless, we added a brief mention of these frameworks also in the introduction (lines 78-86)
Comment R1.3: sample limitations: although the sample size provides useful insights, the recruitment via online survey may have introduced a degree of self-selection bias. It would be valuable to acknowledge this more clearly in the limitations section;
Answer R1.3: corrected as indicated (lines 469-470).
Comment R1.4: demographic variables: if data are available, it could be useful providing more detailed demographic information (e.g., gender, region, rural vs. urban context). This may help readers better understand potential influences on perceptions and bonding;
Answer R1.4: descriptive results have been expanded to increase clarity and transparency at lines 213-221; 227-233. See also answer to comment R3.2
Comment R1.5: interpretation of some findings: the data showing that outdoor-housed chelonians are less likely to be considered family members is compelling and it could be beneficial to deeper discuss possible underlying factors, such as differences in daily interaction, cultural beliefs or environmental constraints;
Answer R1.5: Dear reviewer, thank you for the suggestion. We further explained some of the factors underlying our findings at lines 381-394.
Comment R1.6: clarification of statistical analysis: it is not well clear if the logistic regression model was adjusted for additional covariates or if the reported odds ratio is unadjusted;
Answer R1.6: We thank the reviewer for this helpful comment. In the manuscript, we report the odds ratio for the final logistic regression model obtained through stepwise selection based on the Akaike Information Criterion (AIC). To clarify:
- The adjusted odds ratio reported in the Results section (OR = 4.90, 95% CI = 1.34–23.41) refers to the final model, which included only one predictor — housing environment (outdoor vs. indoor/mixed). No other covariates improved model fit or were retained after AIC-based model selection.
- For transparency, we also present unadjusted odds ratios for all candidate predictors in a separate forest plot (Figure 1 and table 4), allowing readers to evaluate each variable's bivariate association with bond type.
To address this comment, we revised the Results section to explicitly state that the final model retained a single predictor, and that the adjusted odds ratio reflects this model structure. Because no other variables remained, the adjusted and unadjusted odds ratios for ‘environment’ are numerically similar.
Comment R1.7: qualitative data presentation: the qualitative responses are rich and important, anyway including anonymised illustrative quotes in the main text or as supplementary material would support the thematic interpretations and improve reader engagement;
Answer R1.7: Dear reviewer, thank you for your comments. Anonymized illustrative quotes are included in the main text (for example at lines 325-326; 327; 330-331; 336-337; 338-339; 341-342; 352-354); We further assessed the qualitative data and believe that there are no more meaningful quotes to include in the manuscript.
Comment R1.8: methodological transparency: a clearer explanation of the qualitative analysis process, including coding strategy, whether inductive or deductive and whether inter-coder reliability was assessed, would enhance the study’s rigour;
Answer R1.8: Dear reviewer, thank you for your comment. An explanation of the qualitative method was included in the manuscripts. We have now clarified further on lines 181-188.
Comment R1.9: terminology consistency: the decision to refer to pet keepers as “guardians,” which is a thoughtful and ethically loaded choice, may be improved to briefly explain this terminology and ensure consistent use throughout the manuscript.
Answer R1.9: Dear reviewer, thank you for this comment, we now added at lines 115-119 an explanation of why we preferred using the term guardian instead of owner and why sometimes we believe it was necessary to still use the term ownership instead of guardianship.
Overall, this is a well-designed and insightful study that brings attention to an under-researched aspect of human–animal relationships. I believe the manuscript has the potential to make a valuable contribution to the field once the above points are addressed.
Reviewer 2 Report
Comments and Suggestions for Authors
Thank you for considering me as a reviewer of this interesting manuscript. It deals with the human-chelonian bond in certain regions of Italy. The paper is well written, and I have very few remarks, considering that it can be published after addressing them:
- Line 76: close the parenthesis.
- Observation: I think it would be good to indicate if the Chelonian species cited by the respondents are autochthonous or non-native to Italy and if they are legally on sale in the country or authorized to be kept in captivity by the citizens.
Author Response
Comment R2.1: Line 76: close the parenthesis.
Answer R2.1: done
Comment R2.2: I think it would be good to indicate if the Chelonian species cited by the respondents are autochthonous or non-native to Italy and if they are legally on sale in the country or authorized to be kept in captivity by the citizens.
Answer R2.2: We thank the reviewer for this helpful suggestion. In response, we have added information to clarify the native or non-native status of the chelonian species reported by participants, as well as their legal status in Italy regarding sale and private ownership (lines 215-221). Specifically, we note that Testudo hermanni is the only autochthonous (native) species in Italy, while the other terrestrial (Testudo graeca, T. marginata, T. horsfieldii, T. elegans) and aquatic (Trachemys, Graptemys, Sternotherus) species are non-native. The legal status of these species varies: while Testudo hermanni and other Testudo spp. are commonly kept under CITES regulation (Annex A or B), the commercial trade of Trachemys scripta is currently banned in Italy due to its invasive status (Regulation (EU) 1143/2014). We have clarified this in the main text.
Reviewer 3 Report
Comments and Suggestions for Authors
See report attached

Author Response
Human-Chelonian Bond in Italy: An exploratory study of pet turtle and tortoise ownership.
This is an interesting study on the relationship between pet turtles and tortoises, and their owners. I commend the authors in incorporating both survey and qualitative results in one manuscript, to provides a detail understanding in the relationships.
This manuscript well written and structured. However, there are a few statistical issues that the authors need to address before it is suitable for publication.
Major issues
Material & Methods
However, some areas that require a bit more detail and rationale in conducting analysis.
Comment R3.1: The authors need to expand on how the participants were recruited into the study. In particularly the students and clients of a veterinary clinic. How were they invited to participate? Was it by email or flyers? Why only students and not the professional and academic stuff of the Department? Why was only one veterinary clinic used? Did you approach other clinics to be part of the study? Do you know roughly the number of students and clients were approached? It will be ideal if the authors follow and complete the STOBE Checklist for cross-sectional studies https://www.strobe-statement.org/. These checklists ensures the correct information is included in the manuscript.
Answer R3.1: Dear reviewer, thank you for your comment, we have included additional information in the manuscript on lines 129-133 to clarify the recruitment procedures. The survey was disseminated through multiple channels, including social media pages and groups dedicated to reptile enthusiasts, as well as internal university communication platforms. Additionally at lines 205-206 we highlighted that most of the replies (88%) come from social media pages or groups dedicated to reptiles. Although the survey was available for every person involved within the department of veterinary sciences, only 3 students replied to the questionnaire.
Regarding the veterinary clinic, it was the only one involved because it was the only clinic specialized in exotic animals in the surroundings of the university.
Comment R3.2: My biggest concern is conducting a logistic regression on a relatively small sample. Usually, a rule of thumb in conducting any regression models is having at least 50 respondents/records per variable. So, it is not surprising that only one variable was “significant”. I also not sure why conducting a regression model is necessary. Adjustment for covariates is appropriate when the question is explicitly causal (Hernan 2018), which I can see if not the aim in this study? Presenting descriptive results, as in this study is warranted. I was frustrated that there was not a frequency table with n, percentages and 95% confidence intervals of percentages. (I should note that I could not access the Suppl Mat online) Undertaking a regression model is not always necessary. My I have included a few articles on why this is the case. My suggestion is to present the frequency of family member vs non-family member type bond by various factors, with n, % (95% CI), and discuss the percentage difference. This would be more informative than a model.
Some literature in this area:
Carlin JB, Moreno-Betancur M. On the Uses and Abuses of Regression Models: A Call for Reform of Statistical Practice and Teaching. Stat Med. 2025 Jun;44(13-14):e10244. doi: 10.1002/sim.10244. PMID: 40553044; PMCID: PMC12186762.
Miguel A. Hernán, John Hsu & Brian Healy (2019) A Second Chance to Get Causal Inference Right: A Classification of Data Science Tasks, CHANCE, 32:1, 42-49, DOI: 10.1080/09332480.2019.1579578
Hernán MA. The C-Word: Scientific Euphemisms Do Not Improve Causal Inference From Observational Data. Am J Public Health. 2018 May;108(5):616-619. doi: 10.2105/AJPH.2018.304337 2
Conroy S, Murray EJ. Let the question determine the methods: descriptive epidemiology done right. Br J Cancer. 2020 Oct;123(9):1351-1352. doi: 10.1038/s41416-020-1019-z.
Kaufman, J. S. (2017). Statistics, Adjusted Statistics, and Maladjusted Statistics. American Journal of Law & Medicine, 43(2–3), 193–208. https://doi.org/10.1177/0098858817723659
Westreich D, Greenland S. The table 2 fallacy: presenting and interpreting confounder and modifier coefficients. Am J Epidemiol. 2013 Feb 15;177(4):292-8. doi: 10.1093/aje/kws412.
Igelström E, Craig P, Lewsey J, Lynch J, Pearce A, Katikireddi V. Causal inference and effect estimation using observational data. J Epidemiol Community Health 2022;76:960-966.
Answer R3.2: We thank the reviewer for this thoughtful and methodologically grounded comment. We fully agree that statistical modelling should be applied cautiously in exploratory studies, especially with modest sample sizes, and we appreciate the references provided.
Regarding sample size and model suitability, we acknowledge that our sample of 91 respondents is relatively small. While the reviewer cites a rule of thumb of 50 observations per variable, we note that this threshold is more conservative than what is typically applied in applied social and veterinary research. Logistic regression models are commonly used with a minimum of 10 events per predictor variable (EPV), especially when the analysis is exploratory and conducted with care to avoid overfitting. In our case, the outcome variable (“family member” bond) had 43 events, which we considered appropriate for a limited number of candidate predictors. To reduce model complexity, we applied stepwise AIC-based model selection, collapsed sparse categories, and checked for multicollinearity (VIF < 2). Only one variable (indoor/outdoor environment) was retained in the final model, further limiting model burden.
While we agree that regression modelling is most often warranted when addressing explicitly causal questions, our aim was not to infer causality, but to identify potential associations between guardian, animal, and housing characteristics and the type of bond reported. This was made explicit in the manuscript (Section 2.3), and we took care to interpret our results cautiously, avoiding causal language. We used regression modelling as a tool to explore whether any factors stood out in the context of an otherwise descriptive study, and not to assert adjusted or mechanistic effects.
Regarding significance testing and model interpretation, we share the reviewer’s concerns regarding the limitations of relying solely on p-values. In line with recent statistical recommendations (e.g., Wasserstein & Lazar 2016), we already presented effect sizes and 95% confidence intervals alongside p-values in Table 3 and Figure 1. We have revised the manuscript to further de-emphasize significance thresholds and ensure that the interpretation of model results focuses on direction, plausibility, and confidence bounds, rather than binary thresholds.
In order to address the reviewer’s concerns, we improved the manuscript to present detailed descriptive statistics and increase transparency in this exploratory work. In response to the reviewer’s suggestion, we have included an additional table in the main manuscript showing the frequency of “family-member” vs. “non-family-member” bonds across all main variables, including n, percentage, and 95% confidence intervals. This will allow readers to independently evaluate the patterns and support full transparency regarding raw data patterns.
With the (1) expansion of the descriptive results to include n, % and 95% CI for bond types across key demographic and animal-related variables, (2) regression analysis clearly framed as exploratory and interpreted without causal claims, and (3) revised the text to de-emphasize statistical significance, we hope that these adjustments address the reviewer’s concerns and strengthen the manuscript’s clarity and rigour.
Comment R3.3: Another issue with the logistic regression modelling is including/excluding variables based on p values less than 0.05. P values are reliant on sample size, which explains why only one variable is ‘significant’. Although commonly used in scientific literature, misinterpreting study results as real or not based on ‘statistical significance’ determined by an arbitrary p value threshold potentially dismisses genuine effects and can lead to misleading conclusions. I have also included articles on why you should not use significance as a reason for keeping variables or reporting their importance. It is better to present the frequency of family member vs non-family member type bond by various factors.
Literature on using p values:
- Sterne J A C, Cox D R, Smith G D. Sifting the evidence—what's wrong with significance tests? BMJ 2001; 322 :226 doi:10.1136/bmj.322.7280.226
- Wasserstein, R. L., & Lazar, N. A. (2016). The ASA statement on p-values: context, process, and purpose. https://amstat.tandfonline.com/doi/full/10.1080/00031305.2016.1154108
- Greenland, S., Senn, S. J., Rothman, K. J., Carlin, J. B., Poole, C., Goodman, S. N., & Altman, D. G. (2016). Statistical tests, P values, confidence intervals, and power: a guide to misinterpretations. European journal of epidemiology, 31(4), 337-350.
- Amrhein et al. Scientists rise up against statistical significance. Nature 567, 305-307 (2019). doi: 10.1038/d41586-019-00857-9. This article was endorsed by over 800 signatories around the world – we are among those signatories.
- Ronald L. Wasserstein, Allen L. Schirm & Nicole A. Lazar (2019) Moving to a World Beyond “p < 0.05”, The American Statistician, 73:sup1, 1-19, DOI: 10.1080/00031305.2019.1583913).
- Blakeley B. McShane, David Gal, Andrew Gelman, Christian Robert & Jennifer L. Tackett (2019) Abandon Statistical Significance, The American Statistician, 73:sup1, 235-245, DOI: 10.1080/00031305.2018.1527253
Answer R3.3: We thank the reviewer for raising important concerns regarding the limitations of using p-values as thresholds for determining variable inclusion or interpreting findings. We fully agree that over-reliance on statistical significance can be misleading, particularly in small samples, and that binary thresholds (e.g., p < 0.05) do not provide a reliable basis for drawing firm conclusions.
Regarding variable inclusion and model selection, we wish to clarify that we did not include or exclude variables based solely on p-values. Instead, we applied an Akaike Information Criterion (AIC)-based stepwise model selection, which prioritizes model parsimony and penalizes unnecessary complexity. This method considers both fit and simplicity, and is particularly useful in exploratory studies. Furthermore, variables with sparse levels were collapsed based on biological plausibility to ensure robust estimation, and multicollinearity was checked using VIF scores to prevent redundancy. As a result of this cautious approach, only one variable (indoor/outdoor environment) was retained in the final model.
Re- the interpretation of p-values, we acknowledge the limitations of null hypothesis significance testing and share the view that effect sizes and confidence intervals offer a more meaningful interpretation of results. In line with the ASA and Nature editorials cited by the reviewer, we have not treated the p < 0.05 threshold as a strict determinant of “importance” or “truth.” Instead, we reported odds ratios with 95% confidence intervals and clearly framed our findings as exploratory associations, not hypothesis-driven or confirmatory inferences.
To further strengthen the manuscript and respond to the reviewer’s helpful suggestions, we have revised the relevant Results and Discussion sections to:
- Avoid dichotomous “significant/non-significant” language, replacing it with language reflecting confidence, uncertainty, and effect magnitude.
- Emphasize the exploratory nature of the modelling, and clarify that no variable was “excluded” for being non-significant — only for not improving model fit based on AIC.
In line with the reviewer’s suggestion and current recommendations on statistical interpretation, we removed the 'significant/non-significant' legend from Figure 1. The figure now presents unadjusted odds ratios with confidence intervals, encouraging interpretation based on effect size, direction, and estimate precision rather than binary thresholds.
We appreciate the reviewer’s concern and agree that statistical significance should not be the central focus in interpreting results. Our goal is to offer a transparent and cautious interpretation, appropriate to the exploratory nature and sample size of the study.
Minor issues
Material & Methods 3
Comment R3.5: Pg 3, line 107. Did the authors considered other sources in the development of their questionnaires, not just the study conducted by Azevedo et al?
Answer R3.5: We thank the reviewer for this comment. As this study was designed as a replication of the work by Azevedo et al., the questionnaire was intentionally based solely on their original instrument. This was a deliberate methodological choice to preserve comparability between studies and assess whether similar patterns in human–animal bonding could be observed in a different cultural and species context. Replication studies benefit from using the same tools and structure, and we felt that adapting or integrating additional sources would have undermined that goal. We hope the reviewer agrees that methodological consistency is essential in this context.
Results
Comment R3.6: Pg 4, Table 1. Maybe have it so its sorted by most frequent species. This is a suggestion - collapse categories where n=1 into other (Strenotherus spp., …..).
Answer R3.6: Reordered by N, and next alphabetically. Rare species were not collapsed to maintain descriptive richness.
Comment R3.7: Pg 5, Figure 1: Make sure the labels are the same in the written paragraph ie post-laurea; urban/rural. I assume towns is the same as rural?
Answer R3.7: explanation added to methods section. “The variable urbanization was recoded by combining the categories “isolated” and “village” into a single “rural” category, while “town” and “city” were retained as separate levels. For model fitting, “city” served as the reference level.”. Text in section 3.1 was harmonized with figure descriptions of category levels.
Comment R3.8: Pg 5, Table 2. The variable acquisition – what is other? And how many responded
Answer R3.8: The “other” category in the acquisition variable includes responses such as “adopted” and “found,” which were collapsed due to small cell sizes. Specifically: 4 respondents reported adopted, 1 respondent reported found. These were grouped under other (n = 5) for analysis to ensure model stability. This clarification has been added to the Methods section.
Comment R3.9: Pg 7, lines 205 to 219 – this is better placed in a table.
Answer R3.9: We appreciate the reviewer’s suggestion to present this section as a table. However, we chose a narrative format intentionally, as it allowed us to highlight the nuances of species and bond types in a contextualized manner and draw attention to comparative trends across small subgroups. We feel that the text format offers a more readable and integrated account, consistent with the exploratory nature of the analysis, and avoids including another table, in addition to the already numerous ones.
Comment R3.10: Pg 7, Figure 2. This figure needs to formatted that is suitable for journal publication. The y axis should not exceed 100%, and maybe add ‘n’ values.
Answer R3.10: Figure is now formatted correctly, thanks for the comments
Comment R3.11: Pg 8, Figure 3, Again, y-axis should not exceed 5. This result might be better placed in a table with n values and score values with summary statistics.
Answer R3.11: We change the figure accordingly.
Comment R3.12: Pg 8, Table 4. It was confusing having two different n values in the table – one in the title, and another in the header row. It might be better to have a couple of extra rows in the table, and over ‘Emotional connection’, n=38, 45%, than another row at the bottom, ‘Non-emotional responses’. Use lowercase n, in heading, since this is a sample; remove the % in the body of the table since you have already had it labelled in the header row;
Answer R3.12: Dear reviewer, thanks for the suggestions we now clarified the misunderstanding about the n in the caption of the table.
Discussion
Comment R3.13: Pg 10, line 305-310 - As stated previously, may need to reword this section since using p values to determine the importance.
Answer R3.13: We thank the reviewer for this important observation. We have revised this section to de-emphasize p-values as a marker of importance. The sentence has been reworded to avoid language suggesting variable importance was determined by p-values. This revision aligns with current statistical recommendations and with the reviewer’s earlier suggestions.